# Deciphering individual triticale grain weight patterns: A gaussian mixture model approach

**Bo Hwan Kim[1], Hyeok Kwon[2], Wook Kim**[1] *

1 Department of Plant Biotechnology, Korea University, Seoul, Republic of Korea, 2 Institute of Life Science and Natural Resources, Korea University, Seoul, Republic of Korea

* kwook@korea.ac.kr

**Data Availability Statement:** All relevant data are within the Supporting information files.

**Funding:** This work was supported by "Cooperative Research Program for Agriculture Science & Technology Development (Project Number:

## Abstract

Grain weight is one of the key phenotypic traits in crops, closely related to yield. However, the actual structure of grain weight distribution is often overlooked. In this paper, to analyze the characteristics of grain weight, we interpret the weight distribution and structure of individual grains of triticale (× *Triticosecale* Wittmack) from the perspective of a sum of normal distributions, rather than a single normal distribution, using the Gaussian Mixture Model (GMM). We analyzed the individual grain weight distribution of three triticale cultivars (Gwangyoung, Minpung, Saeyoung) bred in Republic of Korea, cultivated under three different seeding rates (150 kg grains per ha, 225 kg grains per ha, and 300 kg grains per ha), over time from 2 to 5 weeks post-heading. Each distribution was fitted using a GMM and evaluated using the Corrected Akaike Information Criterion (AICc) and Bayesian Information Criterion (BIC). It suggests that the distribution of the grain weight is not a single normal distribution, but rather more closely to the distribution composed of two normal distributions. This is hypothesized to be due to the physiological characteristics of the spikelet of *Poaceae*, including triticale, wheat, rye, and oats. Through these results, we recognize the importance of understanding the distribution structure of data and their physiological traits, which is often overlooked in measuring the characteristics of crops.

## Introduction

The phenotypic selection has served as a foundation for breeding, and has been used effectively for the selection and breeding of numerous crops [1, 2]. While our understanding of genomics and genome editing has grown, it is the expressed phenotype that consequently reflects the effectiveness of the plant breeding or genetic modification approaches [3–5]. Especially, for quantitative traits like yield, which involve many genes, traditional phenotypic selection remains an effective method despite the advancement of molecular breeding techniques [6].

The yield-related phenotypic traits, such as spike per area, grain number per spike, grain weight and thousand grain weight were measured and studied to describe final grain weight [7, 8]. For example, in wheat ((*Triticum aestivum* L., Poaceae), the number of grains per unit area and the average grain weight are key indicators related to yield [9–11]. However, in dealing with these phenotypic data, the real distribution is often neglected, and only mean values such as thousand grain weight, which represents the mean grain weight, are used, since identifying individual seed characteristics requires extensive time and labor [12]. In wheat, a high

PJ015339)" Rural development administration, Republic of Korea, the BK21 FOUR program (Grant No. 4299991014324) funded by National Research Foundation of Korea (NRF), and Korea University Grant. The funders had no role in study design, data collection and analysis, decision to publish, or preparation of the manuscript.

**Competing interests:** The authors have declared that no competing interests exist.

level of variation is observed even within a single genotype, and further, within a single spike. This occurs because, in spikelet development and differentiation, the process begins in the middle of the spike and gradually extends towards both ends [7]. Consequently, this leads to variation in grain weight among the apical, central, and basal regions of the spike [13]. Additionally, within the grass tribe Triticeae, including wheat, each floret presents within the spikelet that makes up the spike possesses the potential to develop into a seed. In hexaploid wheat, typically four florets develop into grains, whereas in rye, two florets develop into grains and in barley (*Hordeum vulgare*), one floret develops into one grain [14]. The distribution of wheat grain weight varies depending on which floret the grain was formed in, with grains developed from the second floret in wheat typically being heavier [7, 15]. Conversely, in oat (*Avena sativa* L., Poaceae) the distribution of grain weight often exhibits a bimodal form due to differences between primary and secondary grains in one spikelet, with grains developed in the primary spikelet being heavier than those from the secondary spikelet [16, 17]. These examples suggest that relying solely on simple average values, without considering the distribution of individual grain weights, can lead to erroneous interpretations when comparing and analyzing the characteristics of crops with specific distributions that are not normal distributions.

The triticale (× *Triticosecale* Wittmack), which developed from the hybridization of wheat (*Triticum* spp.) and rye (*Secale cereale* (L.) M.Bieb., Poaceae). It is primarily produced in Europe, and its production has been steadily increasing each year due to its value as a feed grain [18, 19]. Triticale contains higher levels of protein and lysine than wheat and also serves as a good source of vitamin B [18, 20] and it demonstrates a notably higher yield when compared to other grains used for forage, indicating its suitability as a forage crop [19, 21]. Moreover, additional research has been conducted on its use as food and biofuel [22–24].

In analyzing existing grain weight distributions, studies have modeled the weight distribution of rice grains as a sum of gamma distributions [25]. In this study, we utilize the Gaussian Mixture Model (GMM), a probabilistic model in statistics that assumes data is generated from a mixture of several Gaussian (normal) distributions. This allows complex data distributions to be described as a combination of multiple simple Gaussian distributions [26]. GMM is used for analyzing and modeling various distributions, such as audio distribution analysis. In the field of agriculture, it is applied to distinguish between winter crops and other crops or to monitor crop damage caused by floods [27–29]. In this study, we aim to analyze and quantify the real distribution of individual triticale grain weights sampled over time from three cultivars under three different cultivation conditions using Gaussian mixture model. By utilizing the corrected Akaike information criterion (AICc) and Bayesian information criterion (BIC) to determine model fitness [30, 31], we also analyze whether there are specific distribution forms shared among the cultivars of triticale and whether there are different characteristics within these shared distribution forms depending on the cultivar.

## Materials and methods

### Filed experiment and measuring triticale grains

The three cultivar of x *Triticosecale* Wittmack, Gwnagyoung (GW), Minpung (MP) and Saeyoung (SY) were received from Rural Development Administration in Korea, 2021. All three cultivars were developed through varieties introduced from CIMMYT (International Maize and Wheat Improvement Center). Gwangyoung was developed in 2018 through the crossbreed of CTSS93Y00058S-5Y-0Y-0B and CTSS92B00380S-8Y-0Y-0B, Minpung was developed in 2016 through the crossbreed of CTSS92B00968F-B-4M-1Y-0Y-0B and CTSS92B00380S-8Y-0Y-0B, and Seyoung was developed in 2012 through the crossbreed of

ASAD\*2/JUN//ANOAS-5/3/SONNI_6/4/ASAD/ELK54//ERIZO_10 and ERIZO_7/
BAGAL_2//FARAS_1 by National Institute of Crop Science, Korea [32–34].

These seeds were sown at Yeongjung-myeon, Pocheon-si, Gyeonggi-do in republic of
Korea (38.0˚N, 127.2˚E), on October 11, 2022. Referring to the wheat cultivation standards of
the Rural Development Administration of Korea (120 kg N/ha, 150 kg seed per ha) [35–37],
fertilization was done at a rate of 120 kg N/ ha, and the sowing was carried out at three differ-
ent seeding rates: 150 kg seed per ha, 225 kg seed per ha, and 300 kg seed per ha. It was culti-
vated by drilling in rows (8 rows 30 cm apart), and the area of each plot for the treatment
group was set to 2.5 m X 3.5 m. Meteorological information can be found in S1 Table, and the
data was obtained from the Korea Meteorological Administration.

After recording the heading date (May 15, 2023), 60 heads were randomly sampled at one-
week intervals starting from two weeks after heading (2 WAH) up to the fifth week (5 WAH).
As shown by the grains in Fig 1, it is shown the harvest period starting around 4 WAH. Each
sample was dried using a dry oven at 45˚C for 96 hours. After the drying process was com-
pleted, to derive a consistent distribution pattern even under different cultivation conditions,
the five heaviest heads in each treatment group were selected, the grains were threshed out,
and the weight of the individual grains was measured using an electronic scale (HANSUNG,
Model HS220S) capable of measuring up to 1 mg.

## Gaussian mixture model

A Gaussian Mixture Model (GMM) is a statistical model that forms a complex data distribu-
tion by combining multiple Gaussian (normal) distributions. Each Gaussian distribution is
considered as one component within the model. The GMM assumes that the data originates
from multiple different Gaussian distributions. Each data point is assigned a probability of
belonging to each component distribution. These probabilities are called "responsibilities," and
they represent the degree to which a particular data point belongs to each Gaussian distribu-
tion. The probability density function (PDF) of a GMM is defined as follows:

$$p(x) = \sum\nolimits_{i=1}^{k} \pi_i N(x \mid \mu_i, \Sigma_i)$$

while p(x) represents the probability density of the data point x under the model, k is the
number of Gaussian distributions, or components, in the mixture. The $\pi_i$ denotes the mix-
ture weight of the ith component, indicating the proportion of the ith Gaussian in the over-
all mixture. These weights sum to 1, and N ($x \mid \mu i, \Sigma_i$) is the normal (Gaussian) probability
density function for the ith component, with μi as the mean and $\Sigma_i$ as the covariance
matrix.

## Statistical analysis

All statistical analyses were performed by R (version 4.4.0 "Puppy Cup") and R studio (RStu-
dio 2023.09.1+494). The dataset was imported using 'readxl' package (version 1.4.3), and ini-
tial preprocessing steps such as sorting and grouping were performed using 'dplyr' package
(version 1.1.4). All datasets were analyzed after removing outliers by using the IQR outlier
detection method, with values greater than 1.5 times the interquartile range considered as
outliers [38]. After removing outliers, the number of grains used to plot each distribution
ranged from a minimum of 236 to a maximum of 408, with a total of 11,677 grains
measured.

To find a more suitable model for the individual weight distribution of grain, we com-
pared the fit of the commonly used bell-shaped normal distribution (k = 1) to that of

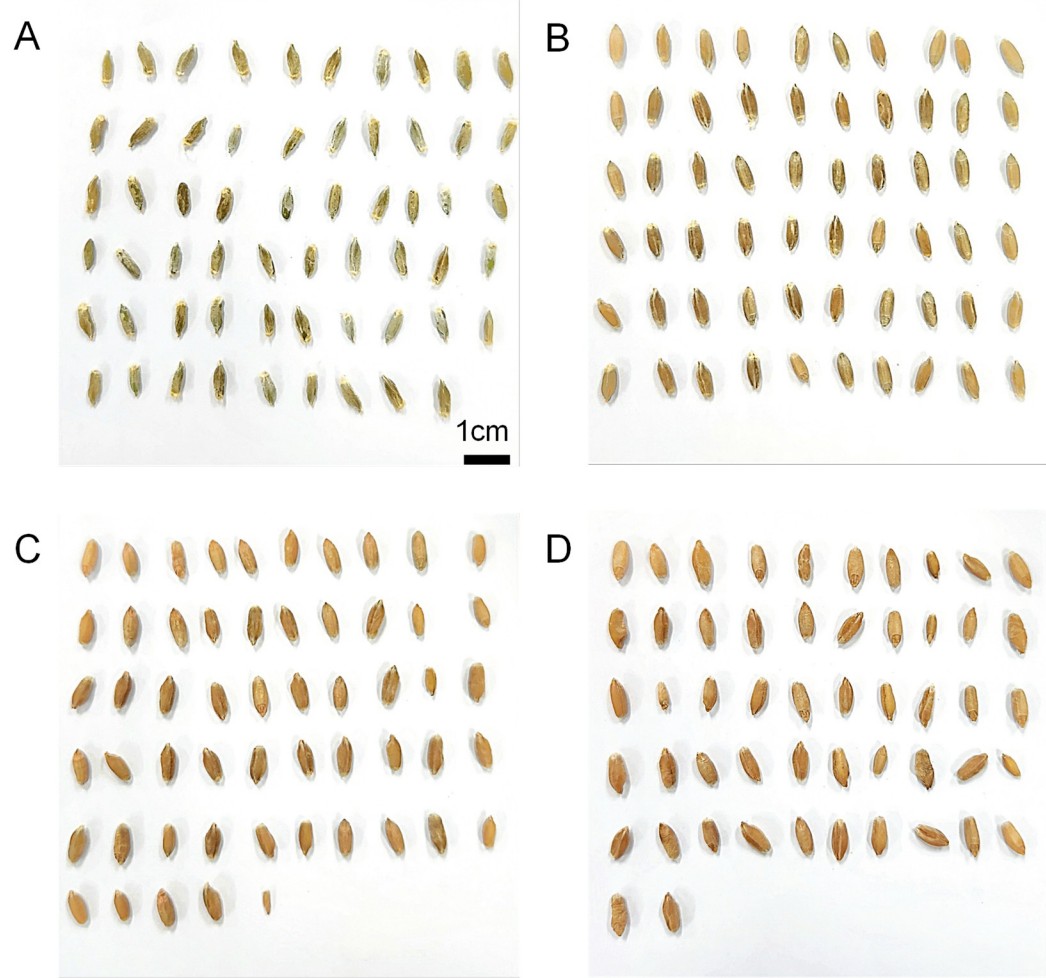

**Fig 1. Differences in triticale grain maturity by developmental stages.** The picture of total triticale (× *Triticosecale* Wittmack) grains from a single head of the Gwangyoung cultivar as an example illustrating the differences in grain development stages. The (A) is 2 weeks after heading (2 WAH), (B) is 3 WAH, (C) is 4 WAH, and (D) is 5 WAH each.

Gaussian Mixture Models (GMM) comprised of the sum of two normal distributions (k = 2) and the sum of three normal distributions (k = 3). The 'normalmixEM' function in 'mixtools' package (version 2.0.0) was used to calculating log-likelihood Gausiaan mixture model (GMM) at k = 2, and 3. For k = 1, the log-likelihood was obtained by fitting a simple normal distribution. Histograms of grain weight distributions were created using 'ggplot2' package (version 3.5.1), with density curves overlaid to represent the fitted GMM components. In the histograms shown in Figs 2 to 4, the data was divided into 30 intervals by setting the bin = 30. Multiple plots were arranged into a single combined plot using 'gridExtra' package (version 2.3). The 'purrr' package (version 1.0.2) facilitated functional programming in R. The map2 function was used to apply functions iteratively over lists of data and parameters, which needed for creating multiple plots efficiently.

For a more precise comparison between the models, we used the corrected Akaike Information Criterion for small sample sizes (AICc), derived from Akaike Information Criterion (AIC) and the Bayesian Information Criterion (BIC). AICc [30] and BIC [31] are defined as

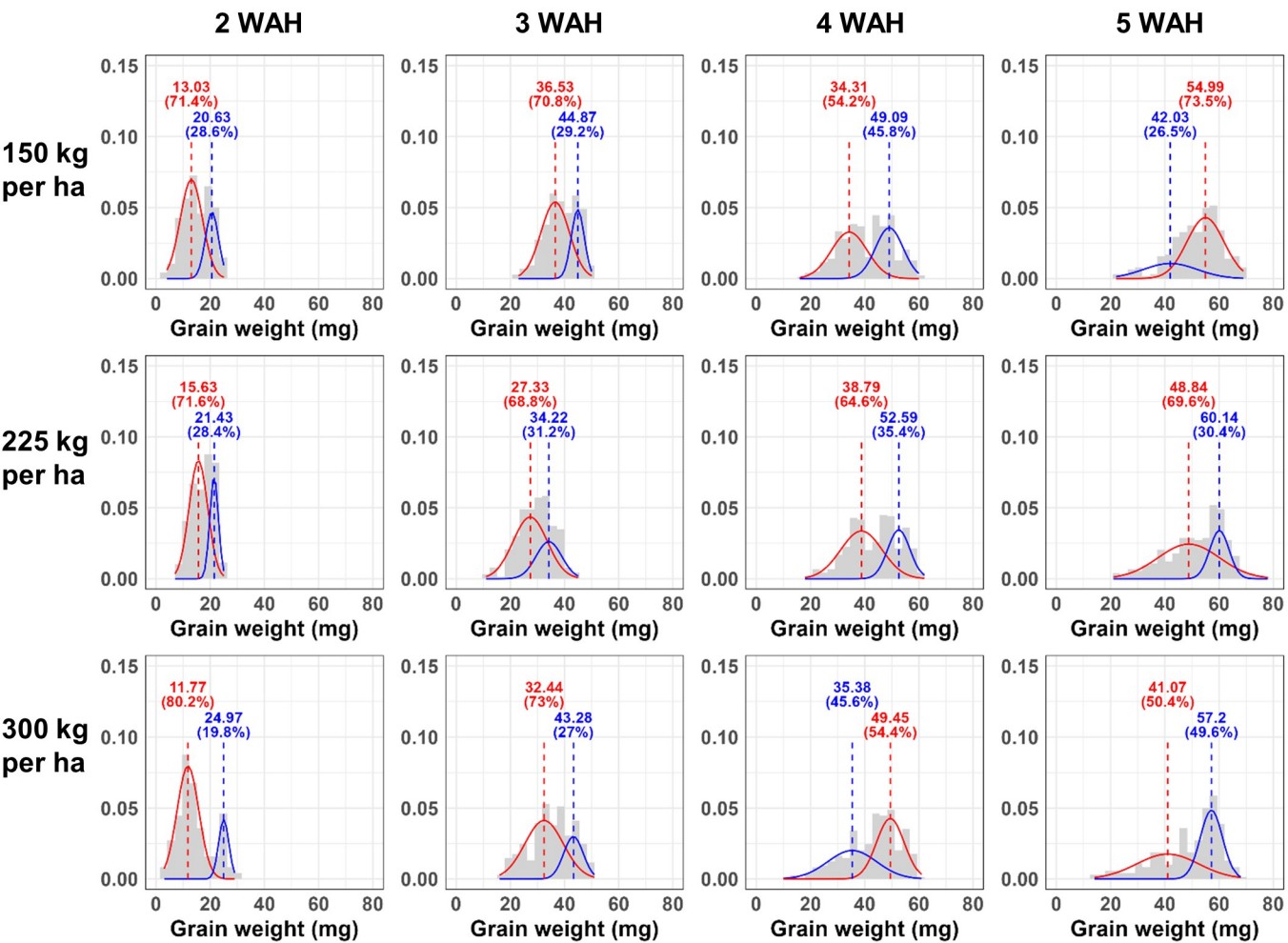

**Fig 2. The Gwangyoung cultivar individual grain weight distribution density plot as a mixture of two normal distributions.** The histogram of individual grain weight of Gwangyoung cultivar by grain developmental stages with Gaussian Mixture Model (GMM) density plot at k = 2 (with two normal distributions). The x axis is individual grain wight of triticale, and the values shown in the graph represent the mean of each distribution, and the percentages values indicate the proportion of each distribution within the overall distribution. The WAH means weeks after heading and 150 kg/ha, 225 kg/ha, 300 kg/ha represents seeding rate per hectare.

follows:

$$AIC = 2K - 2\ln(L) \tag{1}$$

$$AICc = AIC + (2K(K + 1))/(n - K - 1) \tag{2}$$

$$BIC = \ln(n)K - 2\ln(L) \tag{3}$$

In Eq (1), K is the number of parameters in the model and L is the maximum value of the likelihood function for the model. The n is the sample size and K is the number of parameters in the model in (2), and n is the sample size, k is the number of parameters in the model, and L is the maximum value of the likelihood function for the model in (3).

The BIC imposes a greater penalty on models as they become more complex (with an increasing number of parameters) compared to AIC and AICc. For example, the context of

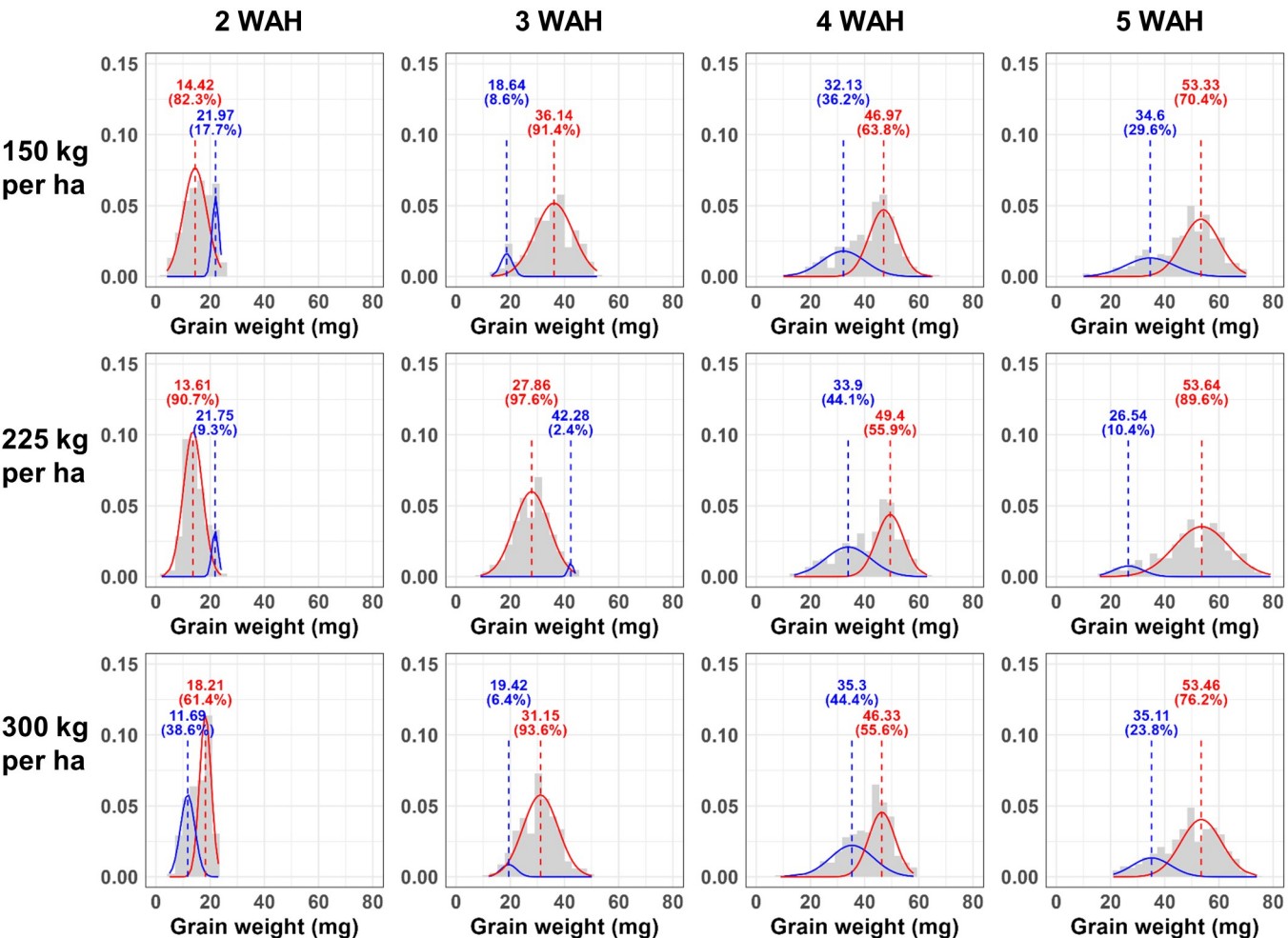

**Fig 3. The Minpung cultivar individual grain weight distribution density plot as a mixture of two normal distributions.** The histogram of individual grain weight of Minpung cultivar by grain developmental stages with Gaussian Mixture Model (GMM) density plot at k = 2 (with two normal distribu-tions). The x axis is individual grain wight of triticale, and the values shown in the graph represent the mean of each distribution, and the percentages values indicate the proportion of each distribution within the overall distribution. The WAH means weeks after heading and 150 kg/ha, 225 kg/ha, 300 kg/ha represents seeding rate per hectare.

model selection criteria such as AIC, AICc, and BIC, an analysis was conducted considering a sample size of n = 300 and varying parameter (k) counts of 2, 5, and 8 across different models. For the AIC complexity penalty, which is computed as 2K, the penalties were 4 for K = 2, 10 for K = 5, and 16 for K = 8. This linear penalty approach reflects the basic principle of AIC in penalizing the number of parameters in a model. The AICc complexity penalty, designed to adjust for finite sample sizes, follows the formula (2), for the given parameter counts, the calculated penalties were approximately 4.040 for K = 2, 10.204 for K = 5, and 16.495 for K = 8. The AICc modification accounts for the sample size, making it a more refined tool for model selection in cases with smaller datasets. Regarding the BIC complexity penalty, which is more sensitive to large sample sizes and follows the formula plog(n), the penalties substantially increased with the number of parameters. The calculated penalties were approximately 11.406 for K = 2, 28.515 for K = 5, and 45.624 for K = 8. The logarithmic nature of the BIC penalty means that it becomes increasingly stringent with larger models, particularly in the context of larger sample sizes.

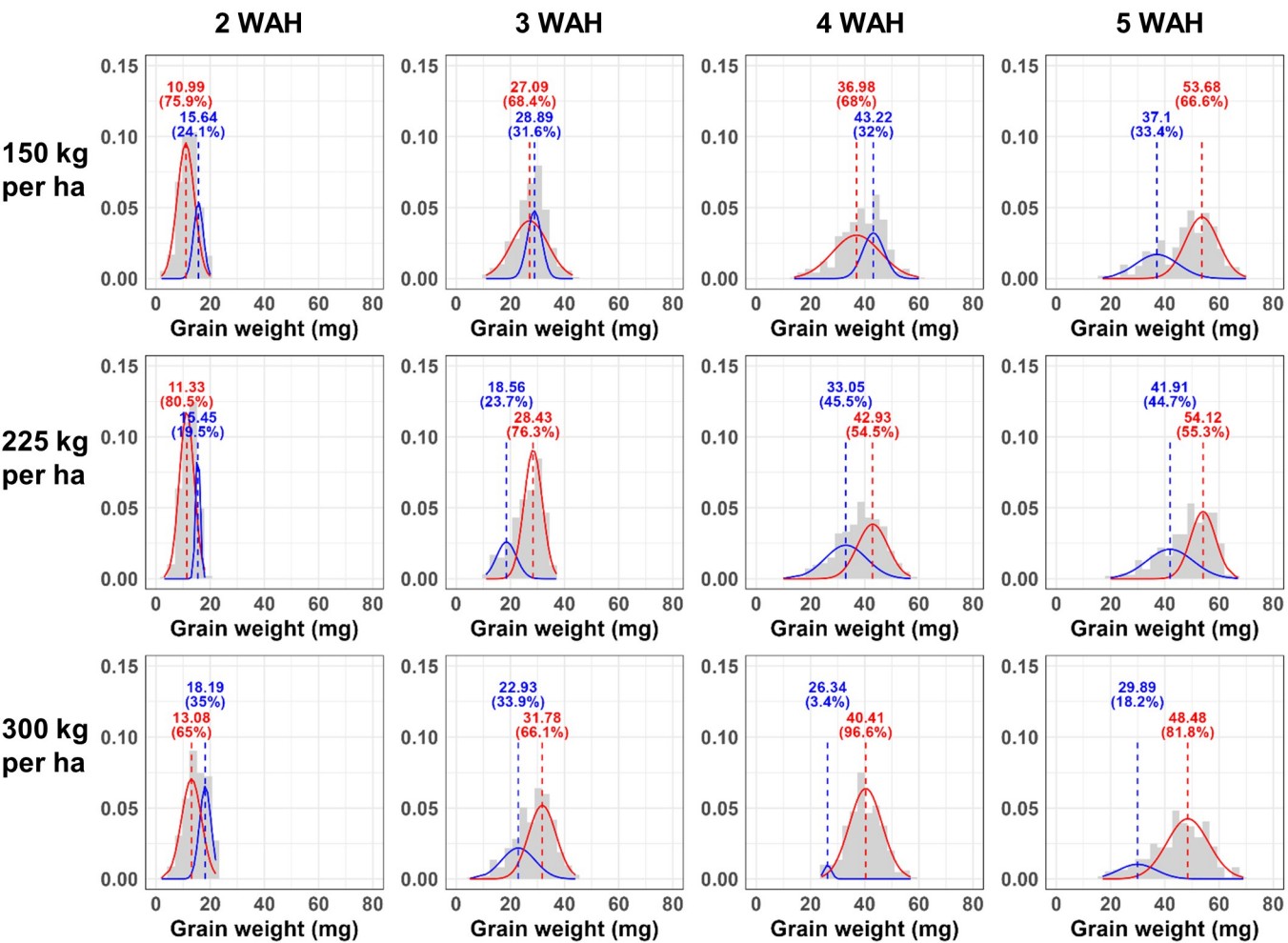

**Fig 4. The Saeyoung cultivar individual grain weight distribution density plot as a mixture of two normal distributions.** The histogram of individual grain weight of Saeyoung cultivar by grain developmental stages with Gaussian Mixture Model (GMM) density plot at k = 2 (with two normal distributions). The x axis is individual grain wight of triticale, and the values shown in the graph represent the mean of each distribution, and the percentages values indicate the proportion of each distribution within the overall distribution. The WAH means weeks after heading and 150 kg/ha, 225 kg/ha, 300 kg/ha represents seeding rate per hectare.

The density plots and the graphs for GMM with k = 2 (Figs 2–4) drawn, and to maintain the distribution shapes and to facilitate comparison between groups, the data for each group were normalized using min-max scaling using 'ggplot2' package in R studio. The shapes of these distributions after Min-Max normalization, which is calculated by (data—min(data)) / (max(data)—min(data)), are depicted in S1–S3 Figs.

## Results

### Individual triticale grain weight

The general characteristics of the individual grains in single head for each cultivar are shown in Table 1 and Fig 1. The mean values are average of individual grain weight from the heaviest five heads out of a total of 60 samples. In Fig 1, the grains sampled 2 weeks after heading are still green. From 2 weeks after heading (2WAH) to 5 weeks after heading (5WAH), the

**Table 1. Summary of individual triticale (x *Triticosecale* Wittmack) grain weight characteristics of three cultivar, Gwangyoung, Minpung, and Saeyoung by developmental stages.**

| Cultivar | Sampling | Seeding rate | Mean [a] | Max | Min | Mean grain number per spike |
|---|---|---|---|---|---|---|
| GW [b] | 2 WAH [c] | 150 kg/ha | 15.1 ± 5.1 gh | 25 | 4 | 68.6 |
| | | 225 kg/ha | 17.2 ± 4.0 g | 25 | 7 | 49.8 |
| | | 300 kg/ha | 14.0 ± 6.1 h | 27 | 3 | 49.4 |
| | 3 WAH | 150 kg/ha | 39.0 ± 6.0 d | 50 | 23 | 53.2 |
| | | 225 kg/ha | 29.4 ± 6.7 f | 45 | 12 | 62.6 |
| | | 300 kg/ha | 35.2 ± 8.1 e | 51 | 13 | 50.0 |
| | 4 WAH | 150 kg/ha | 40.6 ± 9.9 d | 60 | 14 | 57.8 |
| | | 225 kg/ha | 43.3 ± 9.7 c | 62 | 15 | 65.8 |
| | | 300 kg/ha | 43.0 ± 9.9 c | 61 | 15 | 53.6 |
| | 5 WAH | 150 kg/ha | 51.3 ± 9.9 ab | 69 | 23 | 47.2 |
| | | 225 kg/ha | 51.9 ± 11.4 a | 78 | 20 | 53.6 |
| | | 300 kg/ha | 49.0 ± 11.8 b | 68 | 16 | 50.8 |
| MP | 2 WAH | 150 kg/ha | 15.7 ± 4.9 f | 24 | 4 | 71.4 |
| | | 225 kg/ha | 14.3 ± 4.2 f | 24 | 2 | 63.6 |
| | | 300 kg/ha | 15.6 ± 4.0 f | 23 | 4 | 67.4 |
| | 3 WAH | 150 kg/ha | 34.4 ± 8.5 d | 52 | 13 | 63.6 |
| | | 225 kg/ha | 28.0 ± 7.0 e | 44 | 9 | 61.8 |
| | | 300 kg/ha | 30.0 ± 7.4 e | 50 | 10 | 75.2 |
| | 4 WAH | 150 kg/ha | 41.3 ± 10.0 c | 65 | 13 | 72.4 |
| | | 225 kg/ha | 42.3 ± 10.6 c | 63 | 14 | 64.4 |
| | | 300 kg/ha | 41.4 ± 8.4 c | 58 | 17 | 65.0 |
| | 5 WAH | 150 kg/ha | 47.7 ± 11.4 b | 70 | 19 | 68.8 |
| | | 225 kg/ha | 50.8 ± 12.7 a | 82 | 18 | 67.6 |
| | | 300 kg/ha | 48.8 ± 11.1 ab | 74 | 19 | 80.6 |
| SY | 2 WAH | 150 kg/ha | 12.1 ± 3.5 g | 20 | 2 | 80.8 |
| | | 225 kg/ha | 12.1 ± 3.0 g | 18 | 3 | 63.4 |
| | | 300 kg/ha | 14.9 ± 4.0 f | 22 | 5 | 64.2 |
| | 3 WAH | 150 kg/ha | 27.6 ± 5.4 de | 41 | 14 | 75.6 |
| | | 225 kg/ha | 26.0 ± 5.4 e | 37 | 13 | 69.2 |
| | | 300 kg/ha | 28.5 ± 7.1 d | 44 | 11 | 70.8 |
| | 4 WAH | 150 kg/ha | 39.2 ± 7.9 c | 60 | 18 | 81.6 |
| | | 225 kg/ha | 38.3 ± 8.3 c | 57 | 16 | 59.2 |
| | | 300 kg/ha | 39.8 ± 6.7 c | 57 | 22 | 58.4 |
| | 5 WAH | 150 kg/ha | 48.0 ± 10.3 a | 70 | 19 | 73.2 |
| | | 225 kg/ha | 48.7 ± 8.8 a | 67 | 25 | 76.4 |
| | | 300 kg/ha | 45.2 ± 10.3 b | 69 | 18 | 78.4 |

[a] Each mean value represents the average of individual grain weight from the heaviest five heads out of a total of 60 samples. Values are mean ± standard deviation with post-hoc comparison between groups was conducted using ANOVA with Tukey's test, and letters were assigned to indicate significant differences.

[b] GW: Gwangyoung, MP: Minpung, SY: Saeyoung cultivar

[c] WAH stands for Weeks After Heading, indicating the timing of sampling after the heading date

average grain weight gradually increases and separated by other groups by Tukey's post hov in all three cultivars. The average number of Gwangyoung cultivar (GW) grains in a single head was ranged from 47.2 to 68.6, Minpung (MP) was 61.8 to 80.6, and Saeyoung (SY) was 59.2 to 81.6.

## Individual grain weight histogram with GMM and AICc and BIC

The histograms of the individual triticale grain weight with GMM for k = 2 and that of after scaling by the min-max scaling are shown in Figs 2 to 4, and S1 to S3 Figs, respectively. In the histogram, as time progresses, the grains become heavier (the distribution shifts to the right over time) and the shape of the distribution changes from sharp (2 WAH) to broader form (5 WAH). This is also evident from Table 1, where the standard deviation of the mean grain weight increases over time. However, in the density plots after min-max scaling, a similarly left-skewed distribution can be observed across all graphs, regardless the cultivar, seeding rate (150 kg/ha, 225 kg/ha, 300 kg/ha) and when sampled after heading. Moreover, in S2 Table, Shapiro-Wilk normality test result on grain weight and that of scaled represents these distributions do not closely follow a normal distribution. The individual grain weight histograms presented with GMM plots for k = 1 and k = 3 presented in S4 to S9 Figs, respectively.

The values of corrected Akaike Information Criterion (AICc) and Bayesian Information Criterion (BIC) for each model calculated through GMM when k = 1, 2, and 3 are presented from Tables 3–5. The k value means that the actual distribution of triticale grains is composed of one normal distribution (k = 1), a combination of two normal distributions (k = 2), and a combination of three normal distributions (k = 3), respectively. While a higher log-likelihood value indicates a model that is more compatible with the data, reliance on log-likelihood values alone is problematic since they tend to increase as a model becomes more complex. In our experiments, as the complexity of the model increased (as reflected by increasing values of k), the log-likelihood values also increased with one exception (S3 Table). Generally, as the complexity of a model increases, fitness may improve. However, the likelihood of overfitting also escalates. For these reasons, the two information criteria, AICc and BIC used to consider simultaneously the complexity and fitness of the model. Lower values of AICc and BIC indicate a more suitable model. In this experiment, considering the sample sizes ranging from 250 to 408 for each group (Table 1) and 8 parameters for GMM when k = 3, the AICc was used [31, 39].

**Table 2. Representing the most suitable model based on AICc (Corrected Akaike Information Criterion) and BIC (Bayesian Information Criterion) by grain developmental stages of Gwangyoung cultivar.** The AICc and BIC values are from the GMM analysis for Gwangyoung (GW) cultivar harvested in Pocheon, Gyeonggi Province, South Korea in 2023. k = 1 means the assumption that the sample distribution is made up of one normal distribution, k = 2 assumes two normal distributions, and k = 3 assumes three normal distributions.

| | Seeding rate | AICc | | | BIC | | |
|---|---|---|---|---|---|---|---|
| | | k = 1 | k = 2 | k = 3 | k = 1 | k = 2 | k = 3 |
| 2WAH [a] | 150 kg/ha | 2068 | 2056 | **2053** [b] | 2076 | **2075** | 2084 |
| | 225 kg/ha | 1395 | **1379** | 1382 | 1402 | **1396** | 1409 |
| | 300 kg/ha | 1657 | **1585** | 1588 | 1664 | **1603** | 1616 |
| 3WAH | 150 kg/ha | 1708 | **1691** | 1695 | 1715 | **1709** | 1723 |
| | 225 kg/ha | 2068 | 2071 | **2064** | **2075** | 2089 | 2093 |
| | 300 kg/ha | 1736 | 1728 | **1723** | **1743** | 1745 | 1751 |
| 4WAH | 150 kg/ha | 2087 | **2074** | 2077 | 2094 | **2092** | 2106 |
| | 225 kg/ha | 2339 | 2316 | **2313** | 2347 | **2335** | 2343 |
| | 300 kg/ha | 1993 | 1961 | **1961** | 2000 | **1979** | 1989 |
| 5WAH | 150 kg/ha | 1721 | 1707 | **1695** | 1728 | 1724 | **1722** |
| | 225 kg/ha | 2027 | **1998** | 1999 | 2034 | **2016** | 2027 |
| | 300 kg/ha | 1971 | 1898 | **1895** | 1978 | **1915** | 1923 |

[a]) WAH: Weeks After Heading

[b]) The lowest value indicated in bold represents the most suitable

**Table 3. Representing the most suitable model based on AICc (Corrected Akaike Information Criterion) and BIC (Bayesian Information Criterion) by grain developmental stages of Minpung cultivar.** The AICc and BIC values are from the GMM analysis for Minpung (MP) cultivar harvested in Pocheon, Gyeonggi Province, South Korea in 2023. k = 1 means the assumption that the sample distribution is made up of one normal distribution, k = 2 assumes two normal distributions, and k = 3 assumes three normal distributions.

| | Seeding rate | AICc [a] | | | BIC | | |
|---|---|---|---|---|---|---|---|
| | | k = 1 | k = 2 | k = 3 | k = 1 | k = 2 | k = 3 |
| 2WAH [a] | 150 kg/ha | 2123 | 2095 | **2084**[b] | 2131 | **2114** | 2114 |
| | 225 kg/ha | 1802 | **1788** | 1791 | 1810 | **1807** | 1821 |
| | 300 kg/ha | 1875 | **1835** | 1838 | 1883 | **1854** | 1868 |
| 3WAH | 150 kg/ha | 2221 | **2203** | 2211 | 2228 | **2222** | 2241 |
| | 225 kg/ha | **2024** | 2025 | 2028 | **2031** | 2043 | 2057 |
| | 300 kg/ha | **2473** | 2476 | 2481 | **2481** | 2496 | 2512 |
| 4WAH | 150 kg/ha | 2635 | **2594** | 2599 | 2643 | **2613** | 2629 |
| | 225 kg/ha | 2394 | **2351** | 2355 | 2401 | **2370** | 2384 |
| | 300 kg/ha | 2306 | **2281** | 2286 | 2313 | **2300** | 2316 |
| 5WAH | 150 kg/ha | 2599 | **2567** | 2572 | 2607 | **2586** | 2602 |
| | 225 kg/ha | 2662 | **2643** | 2645 | 2670 | **2661** | 2675 |
| | 300 kg/ha | 3033 | **3013** | 3013 | 3041 | **3032** | 3045 |

[a] WAH: Weeks After Heading

[b] The lowest value indicated in bold represents the most suitable

Tables 2–4 present AICc and BIC values for three models (single normal distribution, GMM k = 2, GMM k = 3) across three cultivars (GW, MP, SY). The most appropriate values, which is the lowest, among these have been highlighted in bold. As mentioned above, both the AIC (and its derivative, AICc) and BIC are measures that consider the balance between the fitness and complexity of each model. However, these two criteria are based on fundamentally different assumptions. The AIC is grounded in the assumption that various hypotheses can be

**Table 4. Representing the most suitable model based on AICc (Corrected Akaike Information Criterion) and BIC (Bayesian Information Criterion) by grain developmental stages of Saeyoung cultivar.** The AICc and BIC values are from the GMM analysis for Saeyoung (SY) cultivar harvested in Pocheon, Gyeonggi Province, South Korea in 2023. k = 1 means the assumption that the sample distribution is made up of one normal distribution, k = 2 assumes two normal distributions, and k = 3 assumes three normal distributions.

| | Seeding rate | AICc | | | BIC | | |
|---|---|---|---|---|---|---|---|
| | | k = 1 | k = 2 | k = 3 | k = 1 | k = 2 | k = 3 |
| 2WAH [a] | 150 kg/ha | 2167 | **2160** [b] | 2162 | **2175** | 2180 | 2193 |
| | 225 kg/ha | 1577 | **1553** | 1556 | 1584 | **1572** | 1586 |
| | 300 kg/ha | 1805 | 1793 | **1788** | 1812 | **1812** | 1818 |
| 3WAH | 150 kg/ha | 2463 | 2451 | **2451** | 2471 | **2471** | 2482 |
| | 225 kg/ha | 2136 | 2090 | **2084** | 2143 | **2109** | 2114 |
| | 300 kg/ha | 2322 | **2317** | 2321 | **2329** | 2336 | 2351 |
| 4WAH | 150 kg/ha | 2898 | 2882 | **2882** | 2906 | **2902** | 2914 |
| | 225 kg/ha | 2080 | **2076** | 2079 | **2087** | 2094 | 2108 |
| | 300 kg/ha | 1903 | **1903** | 1907 | **1911** | 1921 | 1936 |
| 5WAH | 150 kg/ha | 2722 | **2691** | 2696 | 2730 | **2711** | 2726 |
| | 225 kg/ha | 2770 | 2719 | **2717** | 2778 | **2738** | 2748 |
| | 300 kg/ha | 2961 | **2944** | 2945 | 2969 | **2963** | 2976 |

[a] WAH: Weeks After Heading

[b] The lowest value indicated in bold represents the most suitable.

wrong to differing degrees, whereas the BIC is based on the assumption that one among several hypotheses could be correct. Researchers may prefer one of these two approaches based on their individual preferences, but consequently, the BIC imposes a greater penalty on the increase of parameters, i.e., the complexity of the model, compared to the AIC [31, 40, 41].

In these Tables 2–4, in all three cultivars, the analysis using GMM with two (k = 2) or three (k = 3) normal distributions consistently represented the lowest AICc values, whereas the simplest model with a single normal distribution (k = 1) or the model with two normal distributions (k = 2) resulted in the lowest BIC values. As an exception, in Table 2, for the GW cultivar at 5 Weeks After Heading (WAH) and a seeding rate of 150kg/ha, both the AICc and BIC showed the lowest values when k = 3. In the case of the MP cultivar, in Table 3, at 3 WAH with a seeding rate of 150kg/ha and 225kg/ha, both criteria indicated the lowest values when k = 1. In the MP cultivar (Table 3), both the AICc and BIC indicate that k = 2 is the most suitable model except those two cases. Generally, as mentioned, it appears that for the AICc, which places more emphasis on model fitness, models with k = 2 and k = 3 are more suitable. Under the BIC criteria, which is stricter about model complexity, models with k = 1 and k = 2 are deemed more appropriate. Moreover, in both criteria, k = 2 was the most frequently selected option and in S2 Table, most of the distributions were rejected for normality by Shapiro-Wilk normality test. Therefore, it can be comprehensively inferred that the distribution of individual grain weights in triticale is more likely constituted by a combination of two normal distributions rather than being represented by a single normal distribution.

Table 5 shows the overall mean of individual grain weight and the results of GMM analysis at k = 2 from the five heaviest heads at 4 and 5 weeks after heading, considered as the harvest time. The total number of grains obtained from 5 heads ranged from 236 to 408. The results of the Tukey post-hoc test in Table 1 showed that all three cultivars represented higher mean values at 5 WAH compared to 4 WAH. Additionally, when the individual grain weight of the three cultivars GW, MP, and SY is analyzed using GMM k = 2 and divided into two normal distributions, differences in the characteristics of the normal distributions are observed according to each cultivar. For GW, in both 4WAH and 5WAH, one of the two normal distributions occupies up to a maximum of 73% of the overall distribution. The MP cultivar, at both 4 WAH and 5 WAH, the distribution with the higher mean constituted a larger share of the total distribution. At 4 WAH, the proportion of the distribution with the higher mean ranged from 56% to 64%, whereas at 5 WAH, it increased to occupy between a minimum of 70% and a maximum of 90%, indicating that the proportion of the distribution with the higher mean increases over time. As depicted in Figs 2–4 and Tables 2–4, SY, at 4 WAH, showed overall distribution form that was closer to a normal distribution, unlike the other two cultivars. However, at 5 WAH, similar to the other cultivars, it was represented that the distribution is divided into a smaller proportion with a lower mean and a larger proportion with a higher mean. The change over time in the difference in means between the two normal distributions derived from GMM analysis also exhibited different patterns according to the cultivar. In the GW cultivar, over time, the distribution with the lower mean exhibited an increase in variance, resulting in a reduction in height. In MP, the distribution with the higher mean showed an increase in variance over time, and the difference in means between the two normal distributions also exhibited a tendency to increase. In SY, no variance change was observed between the 4 WAH and 5 WAH distributions, but an increase in the difference between the two normal distributions was exhibited.

## Discussion

The distribution of individual grain weights varies across different crops. In rice, a bimodal distribution is observed, comprising a small peak near zero and a large main peak [25, 42]. The

**Table 5. The overall mean, mean of 100-grain weight, and GMM (Gaussian Mixture Model) analysis results with k = 2 for each grain from five head 4 and 5 weeks after heading (4 WAH, 5 WAH).** The α and β represent the proportions of each normal distribution within the total distribution in the analysis results when GMM (Gaussian Mixture Model) is applied with k = 2.

| Cultivar[a] | Sampling | Seeding rate | Total Mean[b] | Mean of GMM k = 2 (α)[c] | Mean of GMM k = 2 (β) | Mean difference[d] |
|---|---|---|---|---|---|---|
| GW | 4 WAH | 150 kg/ha | 40.6 | 34.3 ± 6.6, α = 0.54 | 49.1 ± 5.1 β = 0.46 | 14.8 |
| | | 225 kg/ha | 43.3 | 39.0 ± 7.7, α = 0.65 | 52.6 ± 4.1, β = 0.35 | 13.7 |
| | | 300 kg/ha | 43.0 | 35.4 ± 9.1, α = 0.46 | 49.5 ± 5.1, β = 0.54 | 14.1 |
| | 5 WAH | 150 kg/ha | 51.3 | 55.0 ± 6.8, α = 0.73 | 42.0 ± 9.9, β = 0.27 | 13.0 |
| | | 225 kg/ha | 51.9 | 48.8 ± 11.4, α = 0.70 | 60.1 ± 3.6, β = 0.30 | 11.3 |
| | | 300 kg/ha | 49.0 | 41.1 ± 11.4, α = 0.50 | 57.2 ± 4.1, β = 0.50 | 16.1 |
| MP | 4 WAH | 150 kg/ha | 41.3 | 32.1 ± 8.1, α = 0.36 | 47.0 ± 5.4, β = 0.64 | 14.8 |
| | | 225 kg/ha | 42.3 | 33.9 ± 8.4, α = 0.44 | 49.4 ± 5.1, β = 0.56 | 15.5 |
| | | 300 kg/ha | 41.4 | 35.3 ± 8.0, α = 0.44 | 46.3 ± 4.9, β = 0.56 | 11.0 |
| | 5 WAH | 150 kg/ha | 47.7 | 53.3 ± 6.9, α = 0.70 | 34.6 ± 9.0, β = 0.30 | 18.7 |
| | | 225 kg/ha | 50.8 | 53.6 ± 10.1, α = 0.90 | 26.5 ± 5.5, β = 0.10 | 27.1 |
| | | 300 kg/ha | 48.9 | 53.5 ± 7.5, α = 0.76 | 35.1 ± 7.1, β = 0.24 | 18.3 |
| SY | 4 WAH | 150 kg/ha | 39.2 | 37.0 ± 8.9 α = 0.68 | 43.2 ± 4.0, β = 0.32 | 6.2 |
| | | 225 kg/ha | 38.3 | 33.1 ± 7.7, α = 0.46 | 42.9 ± 5.6, β = 0.54 | 9.9 |
| | | 300 kg/ha | 39.8 | 26.3 ± 1.4, α = 0.03 | 40.4 ± 6.0, β = 0.97 | 14.1 |
| | 5 WAH | 150 kg/ha | 48.0 | 53.7 ± 6.1 α = 0.67 | 37.1 ± 7.9, β = 0.33 | 16.6 |
| | | 225 kg/ha | 48.7 | 41.9 ± 8.6, α = 0.45 | 54.1 ± 4.6, β = 0.55 | 12.2 |
| | | 300 kg/ha | 45.2 | 29.9 ± 7.0, α = 0.18 | 48.5 ± 7.7, β = 0.82 | 18.6 |

[a] GW: Gwangyoung, MP: Minpung, SY: Saeyoung cultivar;

[b] Mean of individual grain weight from the heaviest five heads out of a total of 60 samples

[c] Mean ± standard deviation of two distributions and their proportions (α, β) in the overall distribution results from GMM analysis at k = 2.

[d] The difference in means between the two normal distributions derived from the GMM k = 2 analysis results

left-skewed or normal distribution form was shown in wheat and rye grain [12, 43, 44], bimodal in oat [16, 17]. In wheat, differences in the developmental and differentiation processes of spikelet lead to variations in grain weight between the apical, central, and basal regions of the spike, which can influence the distribution of grain weight. Furthermore, within each spikelet, there are typically 2 to 4 florets that have probability hold grains, and grains which originated from second floret were heavier than other grains. For these reasons, wheat grains can be categorized into groups showing different grain weights, which are determined by their specific position on the spike and the floret in which they originated. Generally, wheat grains which from secondary floret are heavier than primary one, thus, it forms the overall distribution by dividing into groups with a lower mean and those with a somewhat higher mean [13, 45]. In rye, which two florets develop into grains, the individual grain weight distribution also appears as a left-skewed distribution with a long tail extending to the left [14, 44]. However, in oats, there is a pronounced bimodal distribution of grain weight, originating from the significant differences between grains derived from primary and those from secondary in spikelet [16, 17]. Triticale, a hybrid of wheat and rye, also possesses more than two florets per spikelet, similar to wheat, and exhibits comparable tendencies [46, 47]. Thus, when predicting and modeling process, it is imperative to simultaneously consider its developmental processes and physiological characteristics.

In the results, as seen in Figs 2 to 4, the distribution pattern of grain weight also changes depending on the grain maturity. While at 2 Weeks After Heading (WAH) the distribution is narrow and peaked, at 3 or 4 WAH it approaches a normal distribution. From 4 WAH to 5

WAH, height of the distribution gradually decreases and spreads broader, exhibiting a left-skewed shape with a longer tail towards the left. For grains harvested at 5 WAH, the analysis using GMM at k = 2 consistently shows one distribution encompassing grains in the 50 mg to 70 mg range, and another distribution composed of grains with smaller weights. Although the proportions vary slightly, this pattern remains. Moreover, in Table 5, the MP cultivar shows the highest average grain weight and proportion of distributions with a high mean at 5 WAH. However, in Table 5, GW and SY exhibit a smaller standard deviation in the distribution with the higher mean compared to MP, indicating a more uniformity. This suggests that distribution characteristics need to be considered in selection or comparisons between cultivars as perspective of uniformity. Traditional representative values, such as the mean, are often inadequate for comparing and analyzing distributions that deviate from normality, such as bimodal distributions. This demonstrates that GMM can be effectively used for comparing and analyzing such distributions. Additionally, in the density plots representing the distribution of individual grain weights after Min-Max scaling normalization (S1 to S3 Figs), it can be observed that the peak of the graph is lowest at the 2 WAH. This shows a different pattern from the graphs before scaling (sharp and narrow at 2 WAH, broadening and lowering peak over time), implying that considering the characteristic of min-max normalization, which preserves the relationships among data values [48], the actual relationships between data values may differ from what is visually apparent, suggesting caution in data interpretation.

In summary, the distribution of the individual grain weight of triticale appears to be composed of two normal distributions, likely due to the structure of the spikelet and the grain formation process rather than just one. This distribution may also be influenced by factors such as harvest time, and cultivar. However, all three cultivars approached harvest time, they represent a long left tail form (left-skewed form), although there were differences in the detailed distribution shape for each cultivar as shown in Figs 2 to 4 and in S1 to S3 Figs, which present density plots using min-max scaling. In fact, even under different cultivation conditions (seeding rate), sampling methods, such as selecting the heaviest spikes for analysis, could consistently derive the characteristics of the distribution that each cultivar possesses, despite variations in cultivation conditions and environment. These results reveal the importance of not only using statistical representative values but also understanding the accurate distribution when depicting not only grain weight, but also the other characteristics of crop. Moreover, as mentioned, variations in grain weight occur depending on which spikelet and which floret the grain developed from. In wheat, differences in grain weight variation are exhibited based on the position within the spike where it formed, and from which numbered floret the grain developed while grains developed from the second floret exhibit the heaviest distribution compared to those developed from other florets [14, 15]. As a result, the total wheat grain weight distribution appears to be left-skewed but closer to unimodal [12]. In the case of triticale, Gaussian Mixture Model (GMM) analysis indicates that the grain weight distribution is better approximated by two normal distributions, suggesting a form closer to bimodal than unimodal. Currently, we are analyzing whether the distribution pattern of individual triticale grain weight is derived from the floret position or from other factors. Therefore, wheat, as one of the major crops that has undergone numerous breeding stages over a long period, might originally have shown a bimodal distribution pattern based on the origin within the spikelet. However, as a result of continuous breeding, they may have inadvertently shifted towards a unimodal distribution. Considering this, in terms of uniformity selection, selection based on grain weight distribution might potentially contribute to enhancing uniformity in the breeding process. Further distribution analysis between wild wheat species and the currently bred cultivars might potentially reveal this implication, enriching our understanding of how selective breeding has influenced the grain weight distribution over time. Measuring the weight of each individual grain and analyzing its distribution itself requires significant time and effort

[12]. For this reason, in the case of thousand grain weight, commonly used to characterize the weight properties of grains, its distribution essentially represents the distribution of sample means across the entire grain population. Hence, due to the Central Limit Theorem, it takes on the shape of a normal distribution, which may differ from the actual distribution of the population, potentially leading to an incomplete understanding of the true characteristics of the distribution. However, the bottleneck of such phenotypic measurements due to requiring time and labor is impeding progress in breeding and selection research [49, 50]. To overcome this, various techniques related to phenotypic research have been employed recently. High-throughput phenotyping (HTP) refers to the application of various technologies and methods to rapidly and accurately measure a wide range of phenotypic characteristics in plants. HTP employs advanced technologies such as automated imaging, remote sensing, robotics, and data analytics to collect detailed information on plant growth, morphology, yield, stress responses, and other important traits [51–53]. Research has been conducted on grains using image scanning techniques to measure various dimensions such as length, width, and area [12, 54, 55] or even quality of the grains [56]. Furthermore, studies are also underway to explore the correlation between data obtained through these imaging techniques and grain weight, which actual yield-related characteristics, as well as research on estimating grain weight [12, 57]. Therefore, if analyses such as GMM are further considered to examine the distribution characteristics of various traits, beyond just individual grain weight, more accurate results can be obtained.

## Conclusion

In this study, we demonstrated that the real distribution of individual Triticale grains is better represented as a mixture of two normal distributions rather than a single normal distribution using GMM, and that this approach allows for quantifiable comparisons of the distributions. Additionally, we identified the shared distribution patterns among the three triticale cultivars and the unique differences between them. we hope that this study will inform that the actual distribution of grain weights could take various forms, and this finding should be considered in phenotyping. Moreover, this GMM analysis method could be used not only for determining the accurate distribution of individual grain weight in triticale but also in other crops and for different characteristics as a new approach.

## Supporting information

**S1 Fig. The density plot of Gwangyoung cultivar with min-max scaled value.** The min-max scaling was calculated by (data—min(data)) / (max(data)—min(data)). The x axis is scaled individual grain weight of triticale. The WAH means weeks after heading and 150 kg/h a, 225 kg/ha, 300 kg/ha represents seeding rate per hectare.
(TIF)

**S2 Fig. The density plot of Minpung cultivar with min-max scaled value.** The min-max scaling was calculated by (data—min(data)) / (max(data)—min(data)). The x axis is scaled individual grain weight of triticale. The WAH means weeks after heading and 150 kg/h a, 225 kg/ha, 300 kg/ha represents seeding rate per hectare.
(TIF)

**S3 Fig. The density plot of Saeyoung cultivar with min-max scaled value.** The min-max scaling was calculated by (data—min(data)) / (max(data)—min(data)). The x axis is scaled individual grain wight of triticale. The WAH means weeks after heading and 150 kg/h a, 225 kg/ha, 300 kg/ha represents seeding rate per hectare.
(TIF)

**S4 Fig. Representing the individual grain weight distribution as a one normal distributions of Gwangyoung cultivar by seeding rate and time.** The histogram of individual grain weight of Gwangyoung cultivar by grain developmental stages with Gaussian Mixture Model (GMM) density plot at k = 1 (with one normal distributions). The x axis is individual grain weight of triticale. The WAH means weeks after heading and 150 kg/h a, 225 kg/ha, 300 kg/ha represents seeding rate per hectare.
(TIF)

**S5 Fig. Representing the individual grain weight distribution as a one normal distribution of Minpung cultivar by seeding rate and time.** The histogram of individual grain weight of Minpung cultivar by grain developmental stages with Gaussian Mixture Model (GMM) density plot at k = 1 (with one normal distributions). The x axis is individual grain weight of triticale. The WAH means weeks after heading and 150 kg/h a, 225 kg/ha, 300 kg/ha represents seeding rate per hectare.
(TIF)

**S6 Fig. Representing the individual grain weight distribution as a one normal distribution of Saeyoung cultivar by seeding rate and time.** The histogram of individual grain weight of Saeyoung cultivar by grain developmental stages with Gaussian Mixture Model (GMM) density plot at k = 1 (with one normal distributions). The x axis is individual grain weight of triticale. The WAH means weeks after heading and 150 kg/h a, 225 kg/ha, 300 kg/ha represents seeding rate per hectare.
(TIF)

**S7 Fig. Representing the individual grain weight distribution as a mixture of three normal distributions of Gwangyoung cultivar by seeding rate and time.** The histogram of individual grain weight of Gwangyoung cultivar by grain developmental stages with Gaussian Mixture Model (GMM) density plot at k = 3 (with three normal distributions). The x axis is individual grain weight of triticale. The WAH means weeks after heading and 150 kg/h a, 225 kg/ha, 300 kg/ha represents seeding rate per hectare.
(TIF)

**S8 Fig. Representing the individual grain weight distribution as a mixture of three normal distributions of Minpung cultivar by seeding rate and time.** The histogram of individual grain weight of Minpung cultivar by grain developmental stages with Gaussian Mixture Model (GMM) density plot at k = 3 (with three normal distributions). The x axis is individual grain weight of triticale. The WAH means weeks after heading and 150 kg/h a, 225 kg/ha, 300 kg/ha represents seeding rate per hectare.
(TIF)

**S9 Fig. Representing the individual grain weight distribution as a mixture of three normal distributions of Saeyoung cultivar by seeding rate and time.** The histogram of individual grain weight of Saeyoung cultivar by grain developmental stages with Gaussian Mixture Model (GMM) density plot at k = 3 (with three normal distributions). The x axis is individual grain weight of triticale. The WAH means weeks after heading and 150 kg/h a, 225 kg/ha, 300 kg/ha represents seeding rate per hectare.
(TIF)

**S1 Table. The temperature and precipitation data for experimental site.** The temperature and precipitation data from the nearest meteorological observation station to the experimental site, sourced from the Korea Meteorological Administration from September 2022 to August

2023.
(DOCX)

**S2 Table. Normality test for each distribution.** Shapiro-Wilk normality test on individual triticale (X *Triticosecale* Wittmack) grain weight.
(DOCX)

**S3 Table. Representing the log-likelihood values by grain developmental stages of Gwan-gyoung (GW), Minpung (MP), and Saeyoung (SY) cultivar.** The log-likelihood values from the GMM analysis for the three cultivars harvested in Pocheon, Gyeonggi Province, South Korea in 2023. The k = 1 means the assumption that the sample distribution is made up of one normal distribution, k = 2 assumes two normal distributions, and k = 3 assumes three normal distributions.
(DOCX)

**S1 File. Raw data on individual triticale grain weights and R code for GMM analysis.**
(ZIP)

## Acknowledgments

The authors thank to other members, Si Ju Kim, Yi Kyeoung Kim, Hey Won Jun, Hyung Gyu Park, and Seung Bin Ki for supporting thins work.

## Author Contributions

**Conceptualization:** Bo Hwan Kim, Wook Kim.

**Data curation:** Bo Hwan Kim.

**Formal analysis:** Bo Hwan Kim.

**Funding acquisition:** Wook Kim.

**Investigation:** Bo Hwan Kim, Hyeok Kwon.

**Methodology:** Bo Hwan Kim, Hyeok Kwon.

**Project administration:** Wook Kim.

**Resources:** Bo Hwan Kim.

**Supervision:** Wook Kim.

**Validation:** Bo Hwan Kim.

**Visualization:** Bo Hwan Kim.

**Writing – original draft:** Bo Hwan Kim, Wook Kim.

**Writing – review & editing:** Bo Hwan Kim.

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
