## [Decision Letter · Decision Letter 0]

5 Jun 2024

PONE-D-24-15581Deciphering Individual Triticale Grain Weight Patterns: A Gaussian Mixture Model ApproachPLOS ONE

Dear Dr. Kim,

Thank you for submitting your manuscript to PLOS ONE. After careful consideration, we feel that it has merit but does not fully meet PLOS ONE’s publication criteria as it currently stands. Therefore, we invite you to submit a revised version of the manuscript that addresses the points raised during the review process.

We look forward to receiving your revised manuscript.

Kind regards,

Somayeh Soltani-Gerdefaramarzi, Ph. D.

Academic Editor

PLOS ONE

2. In the online submission form you indicate that your data is not available for proprietary reasons and have provided a contact point for accessing this data. Please note that your current contact point is a co-author on this manuscript. According to our Data Policy, the contact point must not be an author on the manuscript and must be an institutional contact, ideally not an individual. Please revise your data statement to a non-author institutional point of contact, such as a data access or ethics committee, and send this to us via return email. Please also include contact information for the third party organization, and please include the full citation of where the data can be found.

Reviewers' comments:

Reviewer's Responses to Questions

**Comments to the Author**

1. Is the manuscript technically sound, and do the data support the conclusions?

Reviewer #1: Partly

Reviewer #2: Yes

2. Has the statistical analysis been performed appropriately and rigorously? 

Reviewer #1: Yes

Reviewer #2: Yes

3. Have the authors made all data underlying the findings in their manuscript fully available?

Reviewer #1: Yes

Reviewer #2: Yes

4. Is the manuscript presented in an intelligible fashion and written in standard English?

Reviewer #1: No

Reviewer #2: Yes

5. Review Comments to the Author

Reviewer #1: the wrtitng style of the paper should be checked because there are many unclear sentences and long sentences in the paper.

There no caption for the figures under the figures.

it is not clear what is the application of introducing the suggested distribution for the grain weight.

version of R along with RStudio should be specify separately in the Statistical section.

specify the code that is used for the analysis.

the sampling was began at heading stage, while at this stage there is no filled grain to be weighted. Also, it is not clear for me what is the aim for sampling over time? I could not find its application on the suggested distribution!

specify why different seed density for cultivation has been applied in the experiment.

Is there any differences among your treatment, seed density and cultivars?

do you think that the number of sampled grain is enoguh to suggest a new distribution for grain weight?

the discusion is somewhat weak and it might be better to rewrite it focusing on the application of the suggested method in the apaper.

Reviewer #2: About Manuscript:

Deciphering Individual Triticale Grain Weight Patterns: A Gaussian Mixture Model Approach

In this study, distribution of individual triticale grain weights sampled over time from three cultivars under three different cultivation conditions analyzed using Gaussian mixture model in triticale grains. They observed that the distribution may influenced by factors such as harvest time, cultivar, and sampling methods. The y concluded that The results reveal the importance of not only using statistical representative values but also understanding the accurate distribution when depicting not only grain weight, but also the other characteristics of crop. Overall, it seems that authors try to gather considerable data to address an issue in oat. The results may be usful for improving selection gain in breeding program in future. The writing of the manuscript is also good. Finally, I suggest this manuscript with two comments:

1-The quality of figures would be better.

2- No powerful literature review can be observed in the introduction. I mean, similar research should be review and the novelty of your research compare to them should be mention in the text.

6. PLOS authors have the option to publish the peer review history of their article (what does this mean?). If published, this will include your full peer review and any attached files.

Reviewer #1: **Yes: **

Reviewer #2: No

---

## [Author Response · Author response to Decision Letter 0]

29 Jun 2024

Response to Reviewers

PLOS ONE

Academic Editor

Dear Dr. Soltani-Gerdefaramarzi,

Thank you for the opportunity to revise our manuscript and for the constructive feedback provided by editor and the reviewers. We have carefully addressed each point raised and have made the necessary revisions to enhance the quality and clarity of our manuscript.

We have reviewed the formatting guidelines provided in the PLOS ONE style templates (PDF file) and the submission guidelines at https://journals.plos.org/plosone/s/submission-guidelines#loc-laboratory-protocols. Accordingly, we have updated the title and headings to ensure they meet the capitalization rules and have added a Conclusion section while revising the ending of the Discussion section for coherence. 

Additionally, we have corrected errors in the table values and used the formatting tools at https://pacev2.apexcovantage.com/ to ensure that the figures meet the required format and have renamed the files appropriately. 

We have revised our data availability statement in line with PLOS ONE's Data Policy. We have made the data available in accordance with PLOS ONE's Data Availability policy. We have included the individual grain weight data and the code used for GMM analysis in the supporting information section.

The instances of "data not shown" in our manuscript have been addressed. We have added the relevant data to Supplementary Table 3 to ensure compliance with PLOS ONE's data sharing requirements. All relevant data are now accessible either within the paper or through the supplementary files.

We believe these revisions address the reviewers' concerns and improve the overall quality of our manuscript. Thank you for considering our revised submission. We look forward to your positive response.

Kind regards,

Prof. Dr. Wook Kim

Reviewer 1.

1. the wrtitng style of the paper should be checked because there are many unclear sentences and long sentences in the paper.

We have revised the sentences throughout the paper to improve clarity and shorten overly long sentences.

Line 62 – 65.

Before: The triticale (× Triticosecale Wittmack), which developed from the hybridization of wheat (Triticum spp.) and rye (Secale cereale (L.) M.Bieb., Poaceae), is primarily produced in Europe, and its production has been steadily increasing each year [18, 19].

After: The triticale (× Triticosecale Wittmack), which developed from the hybridization of wheat (Triticum spp.) and rye (Secale cereale (L.) M.Bieb., Poaceae). It is primarily produced in Europe, and its production has been steadily increasing each year due to its value as a feed grain [18, 19].

Line 128 – 130.

Before: The ‘normalmixEM’ function in ‘mixtools’ package was used for calculating log-likelihood Gausiaan mixture model (GMM) at k = 2, and 3. For k = 1, the log-likelihood was obtained by calculating the fit of a simple normal distribution.

After: The ‘normalmixEM’ function in ‘mixtools’ package (version 2.0.0) was used to calculating log-likelihood Gausiaan mixture model (GMM) at k = 2, and 3. For k = 1, the log-likelihood was obtained by fitting a simple normal distribution.

Line 341 - 347.

Before: Moreover, in Table 5, the MP cultivar shows the highest average grain weight and the proportion of distributions with a high average at 5 WAH, but according to GMM analysis results, GW and SY exhibit a smaller standard deviation in the distribution with the higher mean compared to MP, indicating a more uniformity. This suggests that distribution characteristics need to be considered in selection or comparisons between cultivars.

After: Moreover, in Table 5, the MP cultivar shows the highest average grain weight and proportion of distributions with a high mean at 5 WAH. However, in Table 5, GW and SY exhibit a smaller standard deviation in the distribution with the higher mean compared to MP, indicating a more uniformity. This suggests that distribution characteristics need to be considered in selection or comparisons between cultivars as perspective of uniformity. Traditional representative values, such as the mean, are often inadequate for comparing and analyzing distributions that deviate from normality, such as bimodal distributions. This demonstrates that GMM can be effectively used for comparing and analyzing such distributions.

Additionally, we have modified other long and unclear sentences and words throughout the manuscript like removing repetitive phrases such as "as mentioned".

2. There is no caption for the figures under the figures.

According to the journal format, figure captions should not be placed under the figures but in the main text. We have added the captions to the main text.

Line 177 – 180, Line 203 – 222

3. Version of R along with RStudio should be specify separately in the Statistical section.

specify the code that is used for the analysis.

Line 120 to 135.

We have added the versions of R, RStudio, and the packages used, along with the commands used in the analysis, to the Materials and method section. Moreover, we have included the R studio code used for GMM analysis in the supporting information section.

4. The sampling was begun at heading stage, while at this stage there is no filled grain to be weighted. Also, it is not clear for me what is the aim for sampling over time? I could not find its application on the suggested distribution!

specify why different seed density for cultivation has been applied in the experiment.

Is there any differences among your treatment, seed density and cultivars?

the discussion is somewhat weak, and it might be better to rewrite it focusing on the application of the suggested method in the paper.

The goals of this experiment were:

To determine if the grain weight distribution of triticale follows a normal distribution. If not, what distribution does it follow?

To identify whether the grain weight distribution of triticale has a specific form and quantifying it. Moreover, does the form generally remain the same while exhibiting differences depending on the cultivar?

Sampling over time was conducted to verify whether the distribution form changes over time, addressing the experimental objective of identifying if the distribution form has a specific shape. Through min-max scaling in the supplementary figure, we confirmed that samples at 2 WAH and 3 WAH, which are difficult to observe in the histogram, also show a left-skewed distribution with a long left tail.

Additionally, the GMM analysis results indicated that all three cultivars exhibited a specific left-skewed form. While maintaining this specific form, differences between the three cultivars were also observed.

We apologize for the lack of details regarding the experimental goals. We have revised the introduction and discussion sections to include this information and added information to the discussion section on how the results can be applied. (Introduction section (Line 67 -78), Discussion section (Line 341 - 347, Line 354 - 361), and Conclusion section (Line 391 - 403).

Moreover, we have added information to the discussion section on how the results can be applied.

5. Do you think that the number of sampled grains is enough to suggest a new distribution for grain weight?

Although the number of heads sampled per treatment (five) might seem small, we believe it is sufficient to suggest a new distribution due to the consistency of the results observed.

Thank you for your valuable feedback. We apologize for any confusion caused by the lack of clarity regarding the objectives of the study.

Reviewer 2.

First, we would like to express our gratitude for your valuable comments.

We have addressed the points you raised as follows:

1. The quality of figures could be better.

We have added borders to the figures to make them clearer and have included the means of each distribution and the proportion that each distribution contributes to the overall distribution. This information has been added to the figure legend (Line 203–222).

2. No powerful literature review can be observed in the introduction. Similar research should be reviewed, and the novelty of your research compared to them should be mentioned in the text.

Upon review, we realized the last paragraph of introduction section was indeed lacking. We have added relevant studies and expanded on the explanation of GMM and the objectives of this study. (Line 67 – 78)

Once again, thank you for your thorough review.

Best regards,

---

## [Decision Letter · Decision Letter 1]

6 Sep 2024

PONE-D-24-15581R1Deciphering Individual Triticale Grain Weight Patterns: A Gaussian Mixture Model ApproachPLOS ONE

Dear Dr. Kim,

Thank you for submitting your manuscript to PLOS ONE. After careful consideration, we feel that it has merit but does not fully meet PLOS ONE’s publication criteria as it currently stands. Therefore, we invite you to submit a revised version of the manuscript that addresses the points raised during the review process.

**According to the reviewer's comments, the article needs minor revision. Please make the corrections requested by the reviewer and resubmit the article.**

We look forward to receiving your revised manuscript.

Kind regards,

Somayeh Soltani-Gerdefaramarzi, Ph. D.

Academic Editor

PLOS ONE

**Journal Requirements:**

Reviewers' comments:

Reviewer's Responses to Questions

**Comments to the Author**

1. If the authors have adequately addressed your comments raised in a previous round of review and you feel that this manuscript is now acceptable for publication, you may indicate that here to bypass the “Comments to the Author” section, enter your conflict of interest statement in the “Confidential to Editor” section, and submit your "Accept" recommendation.

Reviewer #3: (No Response)

2. Is the manuscript technically sound, and do the data support the conclusions?

Reviewer #3: (No Response)

3. Has the statistical analysis been performed appropriately and rigorously? 

Reviewer #3: (No Response)

4. Have the authors made all data underlying the findings in their manuscript fully available?

Reviewer #3: (No Response)

5. Is the manuscript presented in an intelligible fashion and written in standard English?

Reviewer #3: Yes

6. Review Comments to the Author

**Reviewer #3: **The manuscript “Deciphering Individual Triticale Grain Weight Patterns: A Gaussian Mixture Model Approach” present a novel interpretation of weight distribution and structure of individuals grains in triticale (Triticosecale Wittmack) from the perspective of mixture Gaussian distribution. The descriptions are presented appropriately and the design factors are included properly. There are however, few concerns that would like to address in the following that could be explained better to be clarified to those readers interested in this topic:

#comment1- As is indicated in the manuscript, weight of individual grain can be affected both by development stage of the spike (which part of the spike the grain was developed) and by which floret the grain was developed. The authors controlled for the combination of both seeding rate and time intervals of the growth to compare the distribution within each separate combination. However, it is not clear whether the identified mixture distribution is due to floret number or the spike (middle, upper, lower parts) as in the current study their potential effect has been confounded and cannot be separated. Please explain how these inferred distributions could be related to the mentioned factor(s).

#comment2- I am not sure whether the individual weights (proportion of individual grain memberships) inferred from mixture models specially at k values 2 or more could be derived from the candidate model(s). If yes, then what would be the consequence if some individuals could not be classified into specific distribution? Are such individuals being wrongly sampled as they are potential admixtures? Please explain how the proposed methods could deal with these admixed individuals.

#comment3- As mentioned in the manuscript, the number of individuals analyzed are varied among different combinations (ranged from 236 to 408). There must be some problems in defining the bin widths in the histogram of the individual weight values and comparison between different combinations of time*seed rate.

7. PLOS authors have the option to publish the peer review history of their article (what does this mean?). If published, this will include your full peer review and any attached files.

Reviewer #3: Yes: Salar Shaaf

---

## [Author Response · Author response to Decision Letter 1]

15 Oct 2024

Response to Reviewers

PLOS ONE

Academic Editor

Dear Dr. Soltani-Gerdefaramarzi,

Thank you for the opportunity to revise our manuscript and for the constructive feedback provided by editor and the reviewers. We have carefully addressed each point raised and have made the necessary revisions to enhance the quality and clarity of our manuscript.

We have reviewed the formatting guidelines provided in the PLOS ONE style templates (PDF file) and the submission guidelines at https://journals.plos.org/plosone/s/submission-guidelines#loc-laboratory-protocols. Accordingly, we have updated the title and headings to ensure they meet the capitalization rules and have added a Conclusion section while revising the ending of the Discussion section for coherence. 

Additionally, we have corrected errors in the table values and used the formatting tools at https://pacev2.apexcovantage.com/ to ensure that the figures meet the required format and have renamed the files appropriately. 

We have revised our data availability statement in line with PLOS ONE's Data Policy. We have made the data available in accordance with PLOS ONE's Data Availability policy. We have included the individual grain weight data and the code used for GMM analysis in the supporting information section.

The instances of "data not shown" in our manuscript have been addressed. We have added the relevant data to Supplementary Table 3 to ensure compliance with PLOS ONE's data sharing requirements. All relevant data are now accessible either within the paper or through the supplementary files.

We believe these revisions address the reviewers' concerns and improve the overall quality of our manuscript. Thank you for considering our revised submission. We look forward to your positive response.

Kind regards,

Prof. Dr. Wook Kim

Reviewer 1.

1. the wrtitng style of the paper should be checked because there are many unclear sentences and long sentences in the paper.

We have revised the sentences throughout the paper to improve clarity and shorten overly long sentences.

Line 62 – 65.

Before: The triticale (× Triticosecale Wittmack), which developed from the hybridization of wheat (Triticum spp.) and rye (Secale cereale (L.) M.Bieb., Poaceae), is primarily produced in Europe, and its production has been steadily increasing each year [18, 19].

After: The triticale (× Triticosecale Wittmack), which developed from the hybridization of wheat (Triticum spp.) and rye (Secale cereale (L.) M.Bieb., Poaceae). It is primarily produced in Europe, and its production has been steadily increasing each year due to its value as a feed grain [18, 19].

Line 128 – 130.

Before: The ‘normalmixEM’ function in ‘mixtools’ package was used for calculating log-likelihood Gausiaan mixture model (GMM) at k = 2, and 3. For k = 1, the log-likelihood was obtained by calculating the fit of a simple normal distribution.

After: The ‘normalmixEM’ function in ‘mixtools’ package (version 2.0.0) was used to calculating log-likelihood Gausiaan mixture model (GMM) at k = 2, and 3. For k = 1, the log-likelihood was obtained by fitting a simple normal distribution.

Line 341 - 347.

Before: Moreover, in Table 5, the MP cultivar shows the highest average grain weight and the proportion of distributions with a high average at 5 WAH, but according to GMM analysis results, GW and SY exhibit a smaller standard deviation in the distribution with the higher mean compared to MP, indicating a more uniformity. This suggests that distribution characteristics need to be considered in selection or comparisons between cultivars.

After: Moreover, in Table 5, the MP cultivar shows the highest average grain weight and proportion of distributions with a high mean at 5 WAH. However, in Table 5, GW and SY exhibit a smaller standard deviation in the distribution with the higher mean compared to MP, indicating a more uniformity. This suggests that distribution characteristics need to be considered in selection or comparisons between cultivars as perspective of uniformity. Traditional representative values, such as the mean, are often inadequate for comparing and analyzing distributions that deviate from normality, such as bimodal distributions. This demonstrates that GMM can be effectively used for comparing and analyzing such distributions.

Additionally, we have modified other long and unclear sentences and words throughout the manuscript like removing repetitive phrases such as "as mentioned".

2. There is no caption for the figures under the figures.

According to the journal format, figure captions should not be placed under the figures but in the main text. We have added the captions to the main text.

Line 177 – 180, Line 203 – 222

3. Version of R along with RStudio should be specify separately in the Statistical section.

specify the code that is used for the analysis.

Line 120 to 135.

We have added the versions of R, RStudio, and the packages used, along with the commands used in the analysis, to the Materials and method section. Moreover, we have included the R studio code used for GMM analysis in the supporting information section.

4. The sampling was begun at heading stage, while at this stage there is no filled grain to be weighted. Also, it is not clear for me what is the aim for sampling over time? I could not find its application on the suggested distribution!

specify why different seed density for cultivation has been applied in the experiment.

Is there any differences among your treatment, seed density and cultivars?

the discussion is somewhat weak, and it might be better to rewrite it focusing on the application of the suggested method in the paper.

The goals of this experiment were:

To determine if the grain weight distribution of triticale follows a normal distribution. If not, what distribution does it follow?

To identify whether the grain weight distribution of triticale has a specific form and quantifying it. Moreover, does the form generally remain the same while exhibiting differences depending on the cultivar?

Sampling over time was conducted to verify whether the distribution form changes over time, addressing the experimental objective of identifying if the distribution form has a specific shape. Through min-max scaling in the supplementary figure, we confirmed that samples at 2 WAH and 3 WAH, which are difficult to observe in the histogram, also show a left-skewed distribution with a long left tail.

Additionally, the GMM analysis results indicated that all three cultivars exhibited a specific left-skewed form. While maintaining this specific form, differences between the three cultivars were also observed.

We apologize for the lack of details regarding the experimental goals. We have revised the introduction and discussion sections to include this information and added information to the discussion section on how the results can be applied. (Introduction section (Line 67 -78), Discussion section (Line 341 - 347, Line 354 - 361), and Conclusion section (Line 391 - 403).

Moreover, we have added information to the discussion section on how the results can be applied.

5. Do you think that the number of sampled grains is enough to suggest a new distribution for grain weight?

Although the number of heads sampled per treatment (five) might seem small, we believe it is sufficient to suggest a new distribution due to the consistency of the results observed.

Thank you for your valuable feedback. We apologize for any confusion caused by the lack of clarity regarding the objectives of the study.

Reviewer 3.

Comments 1: The manuscript “Deciphering Individual Triticale Grain Weight Patterns: A Gaussian Mixture Model Approach” present a novel interpretation of weight distribution and structure of individuals grains in triticale (Triticosecale Wittmack) from the perspective of mixture Gaussian distribution. The descriptions are presented appropriately and the design factors are included properly. There are however, few concerns that would like to address in the following that could be explained better to be clarified to those readers interested in this topic:

Answer 1: The analysis for this issue is currently in progress. Initially, the grains were measured without considering a specific distribution shape, as the distribution form had not been identified at that time. Currently, we are analyzing this in another sampling dataset. The triticale cultivars used in this analysis mainly produced grains in the 1st and 2nd florets, similar to rye, and occasionally in the 3rd floret. Interestingly, during the analysis of the distribution of grains developed in the 1st and 2nd florets, we observed that each showed a left-skewed distribution.

However, since this analysis is still in progress using a different dataset, it is difficult to mention this in the main text (as per PLOS ONE's guidelines, "data not shown" is not accepted). Additionally, because this is from a different dataset, we believe it would be difficult to include it in the supplementary materials as well.

Therefore, we have added a brief note on the ongoing analysi on line 373 to 374.

Comments 2: I am not sure whether the individual weights (proportion of individual grain memberships) inferred from mixture models specially at k values 2 or more could be derived from the candidate model(s). If yes, then what would be the consequence if some individuals could not be classified into specific distribution? Are such individuals being wrongly sampled as they are potential admixtures? Please explain how the proposed methods could deal with these admixed individuals.

Answer 2: In GMM, all data points always have a probability of belonging to at least one distribution. While there is a possibility that some data may be assigned to the wrong distribution, this is inherent to GMM’s probabilistic nature of assigning data.

Line 108 to 111, We have added an explanation of how the data is assigned in the GMM.

We believe that the primary issue affecting the reliability of the analysis is the presence of outliers. Therefore, before analyzing the dataset, we conducted outlier removal, and since we divided the samples present in this dataset, we consider the analysis to be reliable.

Comments 3: As mentioned in the manuscript, the number of individuals analyzed are varied among different combinations (ranged from 236 to 408). There must be some problems in defining the bin widths in the histogram of the individual weight values and comparison between different combinations of time*seed rate.

Answer 3: In the case of the histogram, the command was used to divide the data into 30 equal bins. In line 134 to 135, we have added that histogram information

As you mentioned, differences in shape could arise due to the different sample sizes; however, there are min-max normalized density graphs presented in Supplementary Figures 1-3, which help to identify these differences. This information has been added to the main text, line 361 to 362 in discussion section.

Thank you once again for your valuable review comments.

---

## [Decision Letter · Decision Letter 2]

4 Nov 2024

Deciphering Individual Triticale Grain Weight Patterns: A Gaussian Mixture Model Approach

PONE-D-24-15581R2

Dear Dr. Kim,

We’re pleased to inform you that your manuscript has been judged scientifically suitable for publication and will be formally accepted for publication once it meets all outstanding technical requirements.

Kind regards,

Somayeh Soltani-Gerdefaramarzi, Ph. D.

Academic Editor

PLOS ONE

Additional Editor Comments (optional):

Reviewers' comments:

Reviewer's Responses to Questions

**Comments to the Author**

1. If the authors have adequately addressed your comments raised in a previous round of review and you feel that this manuscript is now acceptable for publication, you may indicate that here to bypass the “Comments to the Author” section, enter your conflict of interest statement in the “Confidential to Editor” section, and submit your "Accept" recommendation.

Reviewer #3: All comments have been addressed

2. Is the manuscript technically sound, and do the data support the conclusions?

Reviewer #3: Yes

3. Has the statistical analysis been performed appropriately and rigorously? 

Reviewer #3: Yes

4. Have the authors made all data underlying the findings in their manuscript fully available?

Reviewer #3: Yes

5. Is the manuscript presented in an intelligible fashion and written in standard English?

Reviewer #3: Yes

6. Review Comments to the Author

Reviewer #3: I think the manuscript present a novel interpretation of weight distribution and structure of individual grains in triticale (Triticosecale Wittmack) from the perspective of mixture Gaussian distribution. The comments have been addressed by the authors.

7. PLOS authors have the option to publish the peer review history of their article (what does this mean?). If published, this will include your full peer review and any attached files.

Reviewer #3:

---

## [Editor Report · Acceptance letter]

15 Nov 2024

PONE-D-24-15581R2 

PLOS ONE

Dear Dr. Kim, 

I'm pleased to inform you that your manuscript has been deemed suitable for publication in PLOS ONE. Congratulations! Your manuscript is now being handed over to our production team.

Kind regards, 

on behalf of

Dr. Somayeh Soltani-Gerdefaramarzi 

Academic Editor

PLOS ONE